# Protein Nanofibrils from Fava Bean and Its Major Storage Proteins: Formation and Ability to Generate and Stabilise Foams

**DOI:** 10.3390/foods12030521

**Published:** 2023-01-23

**Authors:** Anja Herneke, Christofer Lendel, Saeid Karkehabadi, Jing Lu, Maud Langton

**Affiliations:** 1Department of Molecular Sciences, Swedish University of Agricultural Sciences (SLU), 750 00 Uppsala, Sweden; 2Department of Chemistry, Royal Institute of Technology (KTH), 100 40 Stockholm, Sweden

**Keywords:** plant protein, fava bean, amyloids, legumin, vicilin, 11S, 7S, microscopy, rheology

## Abstract

Protein nanofibrils (PNFs) have potential for use in food applications as texture inducers. This study investigated the formation of PNFs from protein extracted from whole fava bean and from its two major storage proteins, globulin fractions 11S and 7S. PNFs were formed by heating (85 °C) the proteins under acidic conditions (pH 2) for 24 h. Thioflavin T fluorescence and atomic force microscopy techniques were used to investigate PNF formation. The foaming properties (capacity, stability, and half-life) were explored for non-fibrillated and fibrillated protein from fava bean, 11S, and 7S to investigate the texturing ability of PNFs at concentrations of 1 and 10 mg/mL and pH 7. The results showed that all three heat-incubated proteins (fava bean, 11S, and 7S) formed straight semi-flexible PNFs. Some differences in the capacity to form PNFs were observed between the two globulin fractions, with the smaller 7S protein being superior to 11S. The fibrillated protein from fava bean, 11S, and 7S generated more voluminous and more stable foams at 10 mg/mL than the corresponding non-fibrillated protein. However, this ability for fibrillated proteins to improve the foam properties seemed to be concentration-dependent, as at 1 mg/mL, the foams were less stable than those made from the non-fibrillated protein.

## 1. Introduction

One suggested approach to lower the environmental impact of the food sector is to eat more locally produced plant-based proteins [1]. Fava beans can be cultivated in the Scandinavian climate, but are currently used primarily as animal feed [2]. Whole fava bean has a protein content of 19–39%, with the major proteins comprising two globulin fractions, legumin, and vicilin (11S and 7S) [3]. Fava bean has good potential for use as a locally produced protein source for humans in many different climate zones. However, convincing consumers to exchange animal-based products with plant-based alternatives is not always easy. Alternative products mimicking the appearance and texture of animal-based products can encourage consumers to eat more plant-based foods [4].

The ability to stabilise air bubbles is an important feature for the textural appearance of many food applications, such as bakery products, whipped cream, ice cream, and cheese [5]. By studying the foaming properties, conclusions can be drawn about an ingredient’s function as a structure enhancer. Proteins have a long history of being used as foam stabilisers. Surface-active proteins can stabilise foams by decreasing the surface tension and forming thin interfacial films that capture air bubbles [6]. Small and flexible proteins usually have higher foaming capacity than large and rigid proteins due to their superior ability to reduce surface tension.

Protein nanofibrils (PNFs) from sustainable sources are of considerable interest to many researchers in material sciences due to the good mechanical properties of these nano-scale fibrils [7]. These properties have also captured the interest of food scientists, who believe that PNFs can be used to create interesting texture profiles [8]. One of the most widely utilised methods for producing protein nanofibrils from food protein is to heat the protein at relatively high temperatures (70–90 °C) in acidic conditions [8]. These harsh conditions cause the proteins to unfold and hydrolyse into smaller peptides. The low pH makes the peptides positively charged, which generates repulsing forces [9]. Only highly ordered peptide assemblies, usually rich in β-sheets, can resist these repulsing forces and form PNFs. This results in a sample containing a mixture of peptides and mature fibrils.

In an early study by our group, we demonstrated that it is possible to produce PNFs from a broad range of plant-based proteins by exposing the protein to an acid environment and heat [10]. One of the plant-based PNFs characterised was produced from a whole protein isolate extracted from locally produced fava beans. When visualising the PNFs with atomic force microscopy (AFM), the result showed that depending on the protein source, the PNFs varied in morphology (being either curved or straight) and in length (~220–910 nm). The plant-based PNFs were compared with a well-studied animal-based protein (whey). Here, the result showed that whey formed considerably longer PNFs (several µm) than any of the PNFs from plant-based sources. We hypothesised that whey proteins superiority could be due to the protein isolate being very pure and consisting of a majority of two very small proteins (β-lactoglobulin and α-lactalbumin), with a molecular weight of 18.3 and 14.2 kDa [11]. The plant-based globulins investigated were approximately 10–20 times larger [12,13,14,15,16] than the major protein found in whey proteins. The fava bean protein used in this previous study was extracted in-house with isoelectric precipitation, generating a protein consisting of a mixture of the larger 11S globulins with a molecular weight of ~353 kDa and the smaller 7S globulins with a molecular weight of ~150 kDa [12]. By separating the two globular fractions and comparing the PNF-forming ability with the whole fava bean protein isolate, further information can be revealed about the impact of protein size on PNF formation.

Fibrillated protein from whey/β-lactoglobulin [17,18] and 11S from soy [19] have been reported to give higher foam volume and stability than non-fibrillated protein from the same sources. In the studies using fibrillated β-lactoglobulins by Peng et al. (2017) and 11S from soy by Wan et al. (2021), the fibrils were isolated by filtration to remove or separate unconverted peptides. Peng et al. (2017), showed that the PNFs produced from β-lactoglobulin formed more stable foams than the non-fibrillated protein, even at concentrations as low as 1 mg/mL, especially at a pH close to the isoelectric point (IP) of the protein. On the other hand, Wan et al., (2021) showed that isolated PNFs from the fractionated soy-11S globulin formed less stable foams than samples containing mixtures of PNFs and peptides or the peptides alone. The reason for these conflicting results is not fully understood. In a recently published study from our group, we investigated the foaming properties of curved PNFs from mung bean [20]. Here, we found that fibrillated protein formed more stable foams than non-fibrillated protein from the same source at in the pH range of 4–9. When separating the PNFs and peptides, we observed that at a pH that was close to the IP of the original protein and at a concentration of 1 mg/mL, the sample with isolated PNFs formed more stable foams than the sample consisting of separated peptides. There is still a need to further investigate if and how PNF can be used to stabilise foams, which will generate a better understanding about the future of plant-based PNFs as a texturing ingredient in food applications.

The aims of the present study were: (i) to characterise and compare the PNFs generated from fava bean protein isolate and the two major globular fractions (11S and 7S) in fava bean; and (ii) to investigate the foaming properties of these PNFs. The PNFs were detected using thioflavin T (ThT), a fluorescing dye with a specific affinity to the β-sheet-rich structure of PNFs [21], which were visualised by AFM imaging. Foaming properties such as foam capacity, stability, and volume of half-life were investigated for non-fibrillated and fibrillated proteins at two different concentrations (1 and 10 mg/mL) after the adjustment of the pH to 7. The only difference in the preparation of the samples was that the fibrillated proteins had been heated at 85 °C for 24 h. These results will fill a knowledge gap on how PNF morphology and concentration influence foaming properties.

## 2. Materials and Methods

Fava beans (*Vicia faba* L. var. *Gloria*) were kindly provided by the RISE (Research Institutes of Sweden). Hydrochloric acid (HCl) was procured from VWR International (Paris, France), sodium chloride (NaCl) from VWR (Darmstadt, Germany), and thioflavin T (ThT) from Sigma (New Delhi, India).

### 2.1. Extraction of Protein from Fava Bean, 11S, and 7S

#### 2.1.1. Fava Bean Protein Isolate

The whole fava bean protein isolate was produced according to the method previously described by Herneke et al., (2021) [10] with some minor adjustments. In brief, the fava beans were dehulled and milled in an ultra-centrifugal mill with a 500 µm mesh. The flour was dispersed in deionised water and diluted at a ratio of 1:10, and the pH was adjusted to 8.0 using 2 M NaOH (Figure 1a). The mixture was then stirred for 1 h, followed by centrifugation at 3700× *g* for 30 min. The supernatant was collected, the pH was changed to 4.0 using 2 M of HCl and incubated with continuous stirring for 1 h, and the supernatant was then centrifuged at 3700× *g* for 15 min. The pellet was collected and washed in deionised water at a ratio of 1:10 and the pH was adjusted to 4, and it was centrifuged at 3700× *g* for 15 min. The pellet was collected and freeze-dried.

#### 2.1.2. 11S and 7S

The 11S (legumin) and 7S (vicilin) fractions were extracted from whole fava beans using a similar protocol to the one described by Suchkov et al. (1990) [22] with some modifications (Figure 1a). The flour was dispersed in deionised water and diluted at a ratio of 1:10 (*w*/*w*) at 20 °C. The pH of the mixture was adjusted to 8.0 using 2 M of NaOH (Figure 1a). The mixture was then incubated with stirring at room temperature (20 ± 2 °C) for 1 h, followed by centrifugation (Sorval Lynx 4000l, Thermo Scientific, Waltham, MA, USA) at 3700× *g* for 30 min. NaCl at a final concentration of 0.6 M was added to the supernatant, and the pH was adjusted to 4.8 with 1 M of HCl and then centrifuged at 5000× *g* (20 °C, 20 min). Both the supernatant (i) and pellet (ii) were saved.

(i)The supernatant was diluted to double the volume with deionised water and centrifuged at 1000× *g* for 15 min. The supernatant was saved overnight at 4 °C, followed by another round of centrifugation at 1000× *g* (15 min). The obtained supernatant was diluted with cold distilled water to double the volume and centrifuged again at 1000× *g* for 15 min, and the pellet was collected and freeze-dried (7S).(ii)The pellet was dissolved in 1000 mL of 0.6 M NaCl solution and stirred to complete dissolution, followed by centrifugation at 5000× *g* (30 min). The supernatant was diluted with distilled water to double the volume. The solution was then left at room temperature, and the sediment was collected and freeze-dried (11S).

#### 2.1.3. Protein Content

Using the Kjeldahl method and a conversion factor of 5.4 [23], the crude protein content was determined to be 78.0% for FPI, 85.8% for 11S, and 80.0% for 7S.

### 2.2. Size Exclusion Chromatography

A small amount of each protein isolate of 11S and 7S was dissolved in 25 mM of bicine buffer (pH 9.0, 500 mM NaCl). The protein solution was thoroughly vortexed and run through a PD-10 column using the same buffer. Using an Äkta explorer (GE, healthcare) and UV absorbance of A280 nm, 1 mL of the protein solution obtained was loaded onto a Superdex-200 Hiload 16/600 size exclusion column, and the separation started at a flow rate of 1 mL/min.

### 2.3. Preparation of Protein Nanofibrils from Fava Bean, 11S, and 7S Protein Isolate

Each protein was dissolved in 0.01 M of HCl to a final concentration of 50 mg/mL. The samples were centrifuged at 3700× *g* for 15 min and filtered through a 0.45 µm nylon syringe filter (Merck Millipore, Dublin, Ireland). Protein content was estimated by dry weight measurement after drying at 105 °C for 3 h. The concentration was adjusted to 10 mg/mL using 0.01 M of HCl; the pH was adjusted to 2, and the samples were incubated in an oven at 85 °C for 24 h without stirring. After heating, the fibrillated samples were cooled on ice and then stored at 4 °C. PNF detection was performed directly after the fibrillated samples were cooled, and the foaming experiments were conducted within a week after preparation.

### 2.4. Thioflavin T Fluorescence

A thioflavin T (ThT) fluorescence assay was performed according to the method previously described by Herneke et al., (2021) [10]. In brief, a 100 μL sample was mixed with 900 μL of 50 µM ThT working solution in phosphate buffer and incubated at room temperature for 20 min before testing. The fluorescence was measured using a multi-mode microplate reader (Polarstar Omega, BMG Labtech, Germany) at an excitation wavelength of 440 nm and emission wavelength of 480 nm.

### 2.5. Atomic Force Microscopy

An atomic force microscopy analysis was conducted using a Bruker Dimension FastScan instrument operating in fast scan mode. The samples were diluted between 1:50 and 1:500 in 0.01 M of HCl, and a 10 µL aliquot was applied on a freshly cleaved mica surface and dried in air. FastScan B cantilevers (Bruker, tip radius = 5 nm, spring constant = 2 N/m, peak force amplitude = 150 nm) were used for the experiments, and the micrographs were analysed with the Gwyddion 2.48 tool (http://gwyddion.net/, accessed on 29 June 2022). All samples were measured in at least duplicates in several different locations on the mica plate. The surface that was measured was 3 × 3 µm.

### 2.6. Rheological Measurements

Steady shear measurements of fibrillated samples from whole fava bean protein isolate, 11S, and 7S at 10 mg/mL and pH 2 and 7 were carried out using the same method previously described in Herneke et al. (2023) [20]. The viscosity (η) was recorded during the shear rate from 0.1/s to 500/s on a Discovery HR-3 hybrid rheometer (DHR3) (TA Instruments, USA) with a cone plate geometry of 40 mm diameter, cone angle 2°, and 51 µm gap. All measurements were carried out at 25 °C and were repeated at least three times.

The power law model, Equation 1, was used to fit the viscosity result from fava bean, 7S, and 11S at pH 2 and 7.
(1)$$\eta =K{\dot{\gamma }}^{n-1}$$

*η* is the shear viscosity, $\dot{\gamma }$ is the shear rate, *K* is the consistency index, and *n* is the flow behaviour index. The power law parameters were obtained in the analyse function in TRIOS (TA instrument), and none of the parameters obtained had a fitting of the model (R^2^) below 0.89.

### 2.7. Zeta Potential

Measurements of the zeta potential were carried out using a Z-sizer (Malvern Instruments). Each sample was measured in triplicates, maintaining an attenuation between 7 and 9, and the temperature was 25 °C. All samples were measured at a concentration of 1 mg/mL.

### 2.8. Foaming Properties

To investigate the foaming properties, 10 mL aliquots of 1 or 10 mg/mL of each of the non-fibrillated and fibrillated samples (see Section 2.2) were transferred to 50 mL beakers. The pH of the samples was adjusted to 5 (1 mg/mL) and 7 (1 and 10 mg/mL) with 0.1 and 2 M of NaOH. The probe S25N (IKA^®^, Staufen, Germany) was immersed in the solution, the homogeniser (IKA ^®^ Ultra Turrax T25) was started, and the speed was gradually increased from 8000 rpm to 13,500 rpm. The solution was allowed to foam for a total of 5 min, the foam volume (FV) was recorded, and foaming capacity (FC, %) was determined using the equation:FC = (FV_i_/V_i_) × 100(2)
where FV_i_ is the volume of foam at time zero and V_i_ is volume of the protein solution before whipping.

Foam stability (FS) was calculated based on foam volume after 30, 60, 120, and 360 min standing at room temperature using the equation:FS = (FV_t_/FV_i_) × 100(3)
where FV_t_ is the volume of foam at time t.

Foam height was calculated according to the exponential decay law as:H(t) = H(0)exp(–λt)(4)
where H(t) is foam height at time t, H(0) is initial foam height at time t = 0, and λ (lamda) is a decay constant. This exponential relationship can be converted to a linear equation by taking the natural logarithm of foam height ln(H(t) versus time:ln(H(t) = ln(H(0)) − λt(5)

Half-life (t_1/2_) of the foams was then determined using the equation:t_1/2_ = ln(H(t_1/2_) − ln(H(0))/−λ(6)

### 2.9. Confocal Microscopy

The microstructure of foams made from non-fibrillated and fibrillated samples were detected using the method described by Herneke et al. (2023) [20] with minor modifications. A confocal laser scanning microscope (CLSM; Zeiss LSM 780, Jena, Germany) was used, equipping an inverted Zeiss Axio Observer and supersensitive GaASp detector. The protein distribution in the samples of foams created from non-fibrillated and fibrillated FPI, 11S, and 7S was detected by staining with the fluorescence dye ThT (3 mM, dispersed into phosphate-buffered saline 10 mM PBS, pH 6.8). The stained samples were placed on a concave microscope slide for observation. An Argon operated at 488 nm excitation wavelength and emission wavelengths between 500 nm and 530 nm was used to detect ThT fluorescence. All images were acquired using a C-Apochromat 63x oil immersion objective (1.32 NA) at an image resolution of 1024 × 1024 pixels.

## 3. Results and Discussion

### 3.1. Characterisation of Protein and Nanofibrils

Three different protein isolates were investigated: the whole protein from fava bean isolate (FPI) and its major globular fractions 11S and 7S. All three proteins were extracted with isoelectric precipitation with the addition of NaCl to collect protein fractions 11S and 7S (Figure 1a). Size exclusion chromatography confirmed that the whole protein isolate from fava bean (Figure 1b, orange) contained both 11S (arrow at 57.6 mL) and 7S (arrow at 67.4 mL). The protein extracted with a final concentration of 0.3 M NaCl was collected at 58.0 mL (Figure 1b, green), and the protein extracted with a final concentration of 0.15 M NaCl was collected at 67.2 mL (Figure 1b, blue). The elution volumes of 11S and 7S suggest molecular weights of 350 kDa and 150 kDa, respectively, which are almost identical to the previously reported values (355 kDa and 150 kDa, respectively) [12].

When the three protein isolates at pH 2 were heated at 85 °C for 24 h, the whole protein isolate had a much cloudier appearance than the samples containing fibrillated 11S and 7S proteins (Figure 1c). This cloudiness was most likely because the whole protein isolate contained starch, fibres, and ash residues [24].

Thioflavin T (ThT) dye is commonly used to detect amyloid-like protein nanofibrils due to its strongly enhanced fluorescence when bound to β-sheet-rich structures [21]. Figure 2 shows the average fluorescence recorded in the ThT assay for fava bean and fractions 11S and 7S before (Protein) and after incubation at 85 °C (PNFs). The increase in fluorescence after fibrillation was greatest for the 7S fraction, with an average increase of 60%, followed by whole fava bean, with an average increase of 56%. For fraction 11S, no increase was detected after fibrillation. As mentioned above, extracted fava bean protein consists of a mixture of 11S and 7S (see Figure 1b), indicating that the increase in fluorescence after fibrillation of the whole bean sample was probably due to the 7S fraction (Figure 2). A similar lack of increase in fluorescence after fibrillation for fraction 11S has been previously observed by our research group for protein isolates extracted from oat and rapeseed, which mainly consist of 12S globulins [10,14,15]. However, the AFM analyses revealed fibril structures in the samples, indicating that ThT assay cannot be used as a stand-alone detection method for PNFs.

The AFM results confirmed that heated proteins from fava bean, 11S, and 7S at pH 2 were able to form PNFs (Figure 3a–c’). All PNFs that formed from the three protein isolates had a similar morphology with a straight semi-flexible structure. However, fibrillated samples from the 7S protein appeared to have a higher density of PNFs when analysed with AFM. Most PNFs in the fibrillated 7S samples were short (~100–200 nm) but with some longer fibrils co-existing in the same samples. The longest fibril measured was 2.2 µm.

Based on the results from the ThT assay (Figure 2) and AFM analysis (Figure 3), it can be concluded that 7S globulins from fava bean are superior to 11S globulins in forming PNFs. This agrees with earlier findings on PNF formation from the 7S and 11S globular fractions in soybean [25]. Using several different methods such as the ThT assay, Congo read spectroscopy assay, circular dichroism spectroscopy, and SDS-gel separation techniques, that study confirmed that when heated at pH 2, the protein from the soybean 7S fraction more readily formed PNFs than the 11S fraction under the same conditions. The reason for 7S being superior to 11S globulins in forming PNFs is not fully understood. Tang and Wang speculated that it might be because 7S contains more charged amino acids, which hydrolyses into peptides more easily at low pH and high temperature [25]. The 11S globulin structures also have a higher denaturation temperature than the 7S structures, which might contribute to their ability to form PNFs [26]. Unfolding proteins by heating them above their denaturing temperature increases the ability of hydrolysis to occur by exposing previously buried residues [8]. For 11S from fava bean, the thermal denaturation midpoint is reported to lie at 85 °C, while the corresponding denaturation midpoint for 7S is reported to lie at 76.5 °C at low ionic strength (µ = 0.08) [26]. In the present study, PNF formation was investigated at 85 °C, and hence 7S had more optimal conditions for PNF formation, which might have generated a larger population of unfolded and hydrolysed 7S protein compared with 11S protein.

When the pH of the fibrillated samples was increased to 7, the fibrillary structure degraded in all samples (Figure 3d,e). For the fibrillated fava bean protein, the samples mainly contained globular aggregates (Figure 3d). Both 11S and 7S seemed to have some shorter fibrils, which grouped into larger aggregates in some cases (Figure 3e,f). These results show that the PNFs from whole fava bean and fractions 11S and 7S are less stable at pH 7 than at pH 2. Fibrillated proteins from whey, soybean, and mung bean have also been shown to be less stable at pH 7 [20,27,28]. However, the PNFs from those sources did not fragment as much as the fava bean 11S and 7S PNFs, so our results in this regard were unexpected.

The flow consistency index and flow behaviour index (Table 1) for fibrillated fava bean, 11S, and 7S at pH 2 and 7 were obtained by fitting the apparent viscosity (Figure 4a) with the power law equation (see method section).

Fibrillated fava bean protein had the highest apparent viscosity at pH 2 (Figure 4), which was probably due to residues of polysaccharides in the protein isolate [24] generating aggregates (see Figure 1). This was also confirmed with a higher consistency index for fibrillated fava bean than 11S or 7S at pH 2 (Table 1). When the pH was increased from 2 to 7, the viscosity of all fibrillated samples dropped (Figure 4), which was also correlated with a lower consistency index for all samples at pH 7 (Table 1). All samples at both pH 2 and pH 7 had a shear thinning behaviour, which was confirmed with a negative flow behaviour index (Table 1). This was probably because the PNFs were degraded to smaller particles (see Figure 3d–f). In an earlier study by our group investigating the viscosity of PNFs generated from mung bean protein [20], it was found that those PNFs had a distinctly different morphology than the PNFs obtained from fava bean-based proteins in the present study, having a curved structure instead of straight. The apparent viscosity of PNFs generated from mung bean (Appendix A) under the same conditions as applied here (10 mg/mL, pH 2) showed that the fibrillated mung bean protein had around 1.7–6.3 times higher apparent viscosity than the fibrillated protein from fava bean and its globulin fractions. This was also confirmed with the fibrillated mung bean samples having a higher consistency index (Appendix A) than the fibrillated fava bean, 11S, and 7S (0.386 vs. 0.217, 0.087, and 0.076 respectively) at pH 2. Similar differences in the viscosity profile have been reported for curly and straight PNFs produced from β-lactoglobulin, where the curved PNFs had higher viscosity than the straight PNFs at the same concentration and conditions [29]. Z potential measurements of both fibrillated and non-fibrillated proteins varied between 20.3 and 25.5 mV at pH 2, −5.5 and −26.9 mV at pH 5, and −13.8 and −21.3 mV at pH 7 (Figure 4b). The fibrillated samples had a similar Z potential at the respective pH, as earlier observed for fibrillated whey protein [30]. No differences could be observed for the samples containing fibrillated and non-fibrillated proteins from the globular fraction 7S and 11S compared with whole fava bean isolate. Based on this observation, the salt added during the protein extraction of 11S and 7S did not affect the charge of fibrillated or non-fibrillated protein. The higher viscosity for fibrillated samples at pH 2 compared with pH 7 (Figure 4a) might also be correlated to the high Z potential observed at low pH (Figure 4b). An increase in Z potential indicates a higher degree of repulsion between particles, resulting in a larger interaction size of particles, causing the particles to move less freely, and thus increasing viscosity.

### 3.2. Foaming Properties

The foaming properties (foaming capacity, foam stability, half-life) of non-fibrillated (protein) and fibrillated (PNF) fava bean, 11S, and 7S were evaluated over a time interval of 15–360 min at pH 7 (Figure 5a,b). Both the non-fibrillated and fibrillated samples were able to form foams after mixing (Figure 5a). However, the increase in foam volume for the fibrillated fava bean and 7S samples was 177% higher, and the increase for the 11S samples was 102% higher than for the non-fibrillated samples. In addition, the fibrillated samples generated much more stable foams than the non-fibrillated samples (Figure 5b,c). In particular, the non-fibrillated fava bean samples were unstable over time. Interestingly, the foams made from fibrillated 11S protein had the longest half-life among all samples investigated (Figure 5c).

To investigate whether these improved foaming properties persisted at very low concentrations for fibrillated protein, foams produced from non-fibrillated (protein) and fibrillated (PNFs) fava bean, 11S, and 7S proteins were diluted to a final concentration of 1 mg/mL at pH 7 (Figure 6a,b). The foaming capacity was found to be almost identical for all three protein isolates irrespective of whether the samples were fibrillated (Figure 6a). However, the foams produced from fibrillated protein were less stable than those from the corresponding non-fibrillated protein (Figure 6b,c). Interestingly, the foam created from fibrillated 11S protein was the least stable, contradicting the observations made at higher concentrations. This indicates that the ability of the fibrillated proteins to stabilise foams was concentration-dependent.

This was confirmed by confocal microscopy of the foam structure (Figure 7). The non-fibrillated samples seemed to have a more substantial fluorescent film of air bubbles and a more uniform distribution of small and large bubbles. Only some of the peptides formed during incubation at low pH and high temperature contribute to the fibril formation [31]. When investigating the foaming properties of fibrillated 11S protein from soybean as a mixture and when separated into a pure fibril and pure peptide fraction, Wan et al. (2021) concluded that the peptides, and not the PNFs, contributed to the stability of the foams [19]. In contrast to the earlier study by our research group [20], we found that when fibrillated mung bean protein was separated into a pure fibril fraction and a pure peptide fraction, the pure PNFs still formed more stable foams at pH 5 than the peptide fraction at a concentration of 1 mg/mL.

However, it is still not fully understood how curved PNFs contribute to stabilising the foams at low concentrations. A possible explanation is that curved PNFs increase the bulk viscosity in the solution’s continuous phase and thereby sterically hinder bubble rupture. At a pH close to the isoelectric point, the viscosity increased even more due to the aggregation of the PNFs, which probably explains why the foam produced from pure mung bean fibrils was most stable at pH 5 [20]. The fibrillated protein from fava bean, 11S, and 7S did not form stable foams at this low concentration (1 mg/mL), even at pH 5 (Appendix A). The less stable foams probably developed because the straight PNFs generated from fava bean and its major globulin fractions (11S and 7S) were not sufficiently viscous to help stabilise the continuous phase between the bubbles at these low concentrations. Additionally, it cannot be excluded that the peptides in the fibrillated fava bean, 11S, and 7S samples had an impact on the superior foaming properties at the higher concentration (10 mg/mL), as previously observed for fibrillated samples from soy 11S [19]. The PNFs produced from fava bean and its globular fraction might be too inflexible to stabilise the interfacial film of the air bubbles. At the lower concentration (1 mg/mL), the peptides might be too diluted to maintain the stabilising effect observed at higher concentrations. To summarise, based on the results from this study, our previous study about curved PNFs from mung bean, and the study conducted by Wan et al. (2021) about PNF from soy 11S [19], it appears that no general conclusions can be drawn about plant-based PNFs foam stabilisation ability. PNFs will have different morphology (straight/curved) depending on the protein sources. Our findings indicated that curved PNFs are superior in their foam stabilising ability because of their ability to form a more viscous sample.

The non-fibrillated proteins had approximately the same foaming capacity and foam stability/half-life at low concentrations (Figure 6a–c) as at high concentrations (Figure 5a–c). Similarly, a previous study investigating the foaming properties of a fava bean protein isolate at pH 7 and concentrations from 0.1–3% (*w*/*v*) observed that higher concentrations did not give any significant improvement in foaming capacity or stability [32]. In contrast, the foams generated from non-fibrillated protein in the present study had higher foaming capacity. This might be due to the differences in the method used for foam formation or sample preparation. The only difference in treatment between the non-fibrillated and fibrillated proteins examined in this study was that the non-fibrillated proteins were not heated. The initial adjustment of the pH to 2 might have caused the partial unfolding of the native protein structure, generating smaller and more flexible structures and exposing hydrophobic sites that could help stabilise the air/liquid interface [33].

## 4. Conclusions

This study showed that whole protein isolates from fava bean and its two major globulin fractions, 11S and 7S, form straight semi-flexible PNFs at pH 2 when heated at 85 °C for 24 h. Based on the data from the ThT assay and AFM imaging, the 7S fraction forms PNFs more easily than the 11S fraction. The PNFs formed in all fibrillated samples were fragmented when the pH was increased from 2 to 7. At a concentration of 10 mg/mL, the fibrillated protein from all three fractions (whole fava bean, 11S, 7S) formed more voluminous and more stable foams than the non-fibrillated proteins. Fibrillated proteins probably stabilise foams due to their ability to increase the viscosity of the continuous phase, as indicated by the finding that foam stability decreased when diluting the fibrillated proteins to 1 mg/mL, which was not the case for the non-fibrillated proteins. It cannot be excluded that the peptides within the fibrillated samples also contributed to the improved foaming properties at the higher concentration. The results from this study and our earlier study on foaming properties of mung bean-based PNFs show that curved PNFs are superior to straight semi-flexible PNFs in their ability to stabilise foams at low concentrations (1 mg/mL). The results from these studies have generated a greater understanding of how plant-based PNFs with different morphologies can contribute to stabilising future food applications. The introduction of stable air bubbles in food products are important for the appearance and mouthfeel of many food applications. Today, animal proteins are superior to plant-based proteins in the aspect of creating stable bubbles in food. Here, we show that by reconstructing the plant proteins into PNFs, the ability to stabilise air bubbles dramatically increases, generating new insight into how plant protein can be used to create suitable food applications.

## Figures and Tables

**Figure 1 foods-12-00521-f001:**
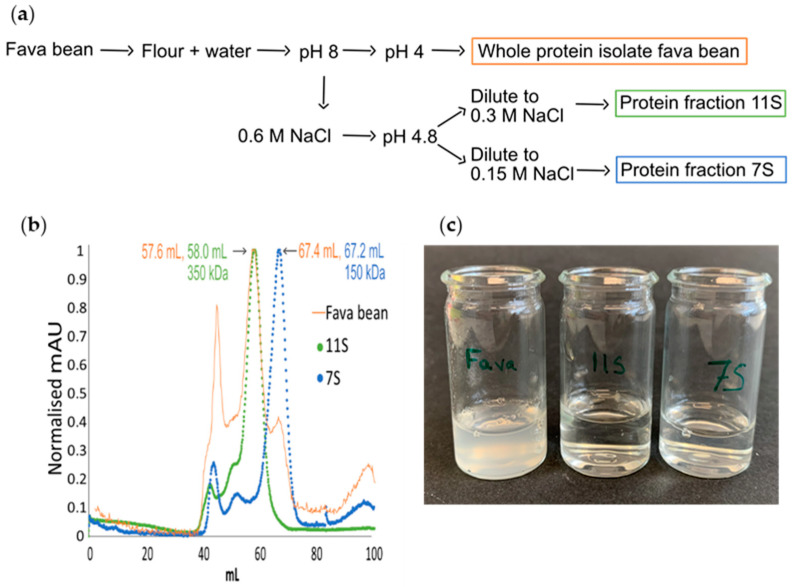
(**a**) Flow chart showing how whole protein isolate and its major globular fractions 11S and 7S were extracted from fava bean using the pH shift method and different concentrations of NaCl. (**b**) Results of normalised size exclusion chromatography of whole protein isolate from fava bean (orange) and fractions 11S (green) and 7S (blue). (**c**) Visual appearance at concentration 10 mg/mL and pH 2 of fibrillated protein from (left to right) fava bean, fraction 11S, and fraction 7S.

**Figure 2 foods-12-00521-f002:**
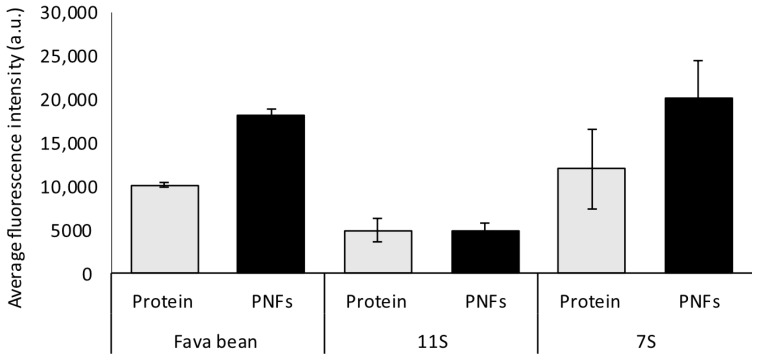
Average fluorescence intensity of non-fibrillated (protein) and fibrillated (protein nanofibrils, PNFs) samples from fava bean protein isolate and fava bean globular fractions 11S and 7S at pH 2 and concentration 10 mg/mL. The error bars refer to the standard deviation.

**Figure 3 foods-12-00521-f003:**
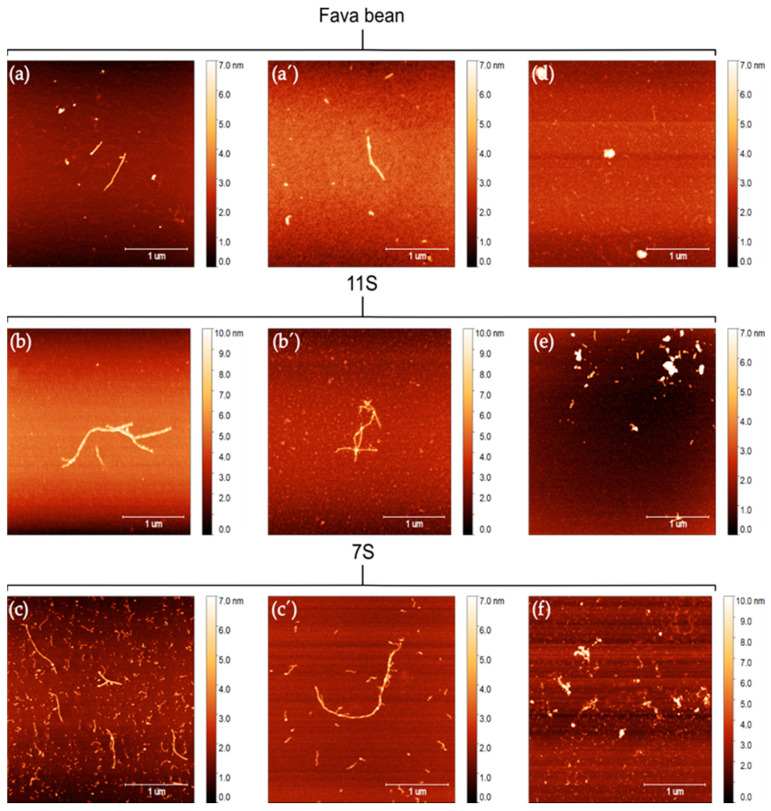
Atomic force microscopy images of two separate batches of fibrillated protein at pH 2 from (**a**–**a’**) fava bean isolate, (**b**–**b’**) fava bean globular fraction 11S, and (**c**–**c’**) fava bean globular fraction 7S. Atomic force micrographs of fibrillated protein from (**d**) fava bean isolate, (**e**) 11S, and (**f**) 7S after pH adjustment to 7.

**Figure 4 foods-12-00521-f004:**
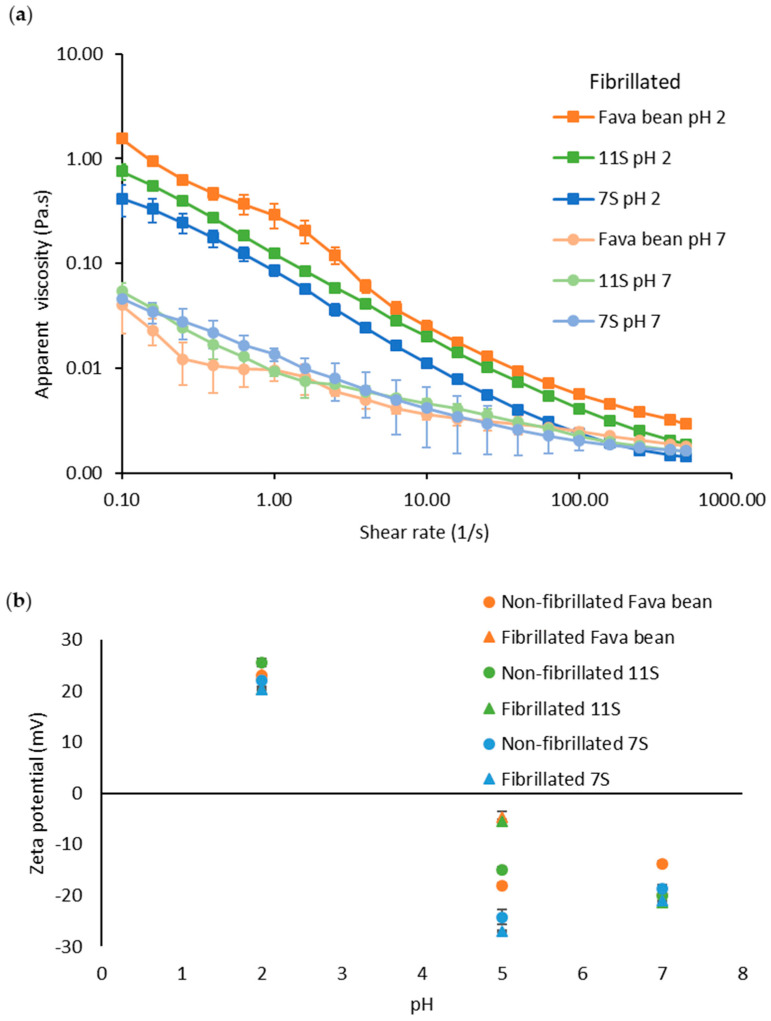
(**a**) Apparent viscosity versus shear rate (0.1–500 s^−1^) for fibrillated protein from fava bean, 11S, and 7S at pH 2 (squares) and pH 7 (circles). All samples were at a concentration of 10 mg/mL. (**b**) Z potential of non-fibrillated and fibrillated fava bean (orange), 11S (green), and 7S (blue) protein at pH 2, 5, and 7. The error bars in figure (**a**,**b**) refer to the standard deviation.

**Figure 5 foods-12-00521-f005:**
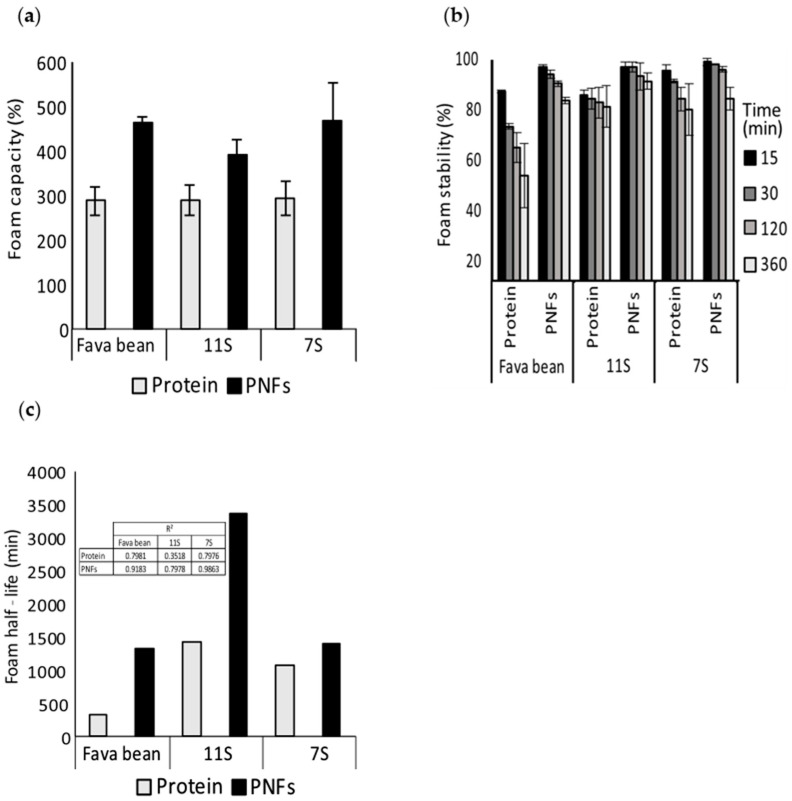
(**a**) Foaming capacity, (**b**) foam stability, and (**c**) foam half-life of non-fibrillated (protein) and fibrillated (protein nanofibrils, PNFs) protein from fava bean and its globular fractions 11S and 7S. The foams were generated at pH 7 and had a concentration of 10 mg/mL. The error bars in figure (**a**,**b**) refer to the standard deviation, and the insert in (**c**) shows the model fit (R^2^).

**Figure 6 foods-12-00521-f006:**
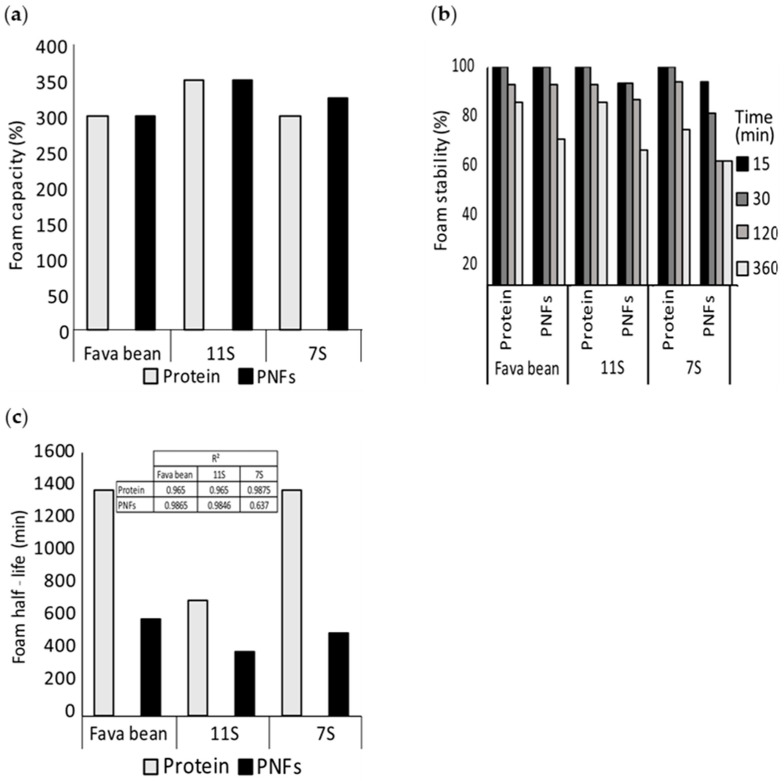
(**a**) Foaming capacity, (**b**) foam stability, and (**c**) foam half-life of fibrillated and non-fibrillated protein from fava bean and its globular fractions 11S and 7S. The foams were generated at pH 7 and concentration of 1 mg/mL. The insert in (**c**) shows the model fit (R^2^).

**Figure 7 foods-12-00521-f007:**
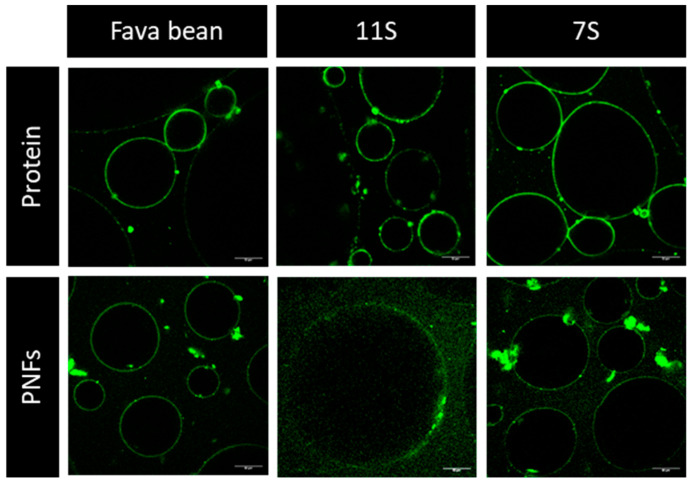
Confocal images (staining with thioflavin T fluorescent dye) of bubbles stabilised by non-fibrillated (protein) and fibrillated (protein nanofibrils, PNFs) proteins from fava bean and its globular fractions 7S and 11S. Scale bar 50 µm.

**Table 1 foods-12-00521-t001:** The flow consistency index (K) and flow behaviour index (n) for fibrillated protein from fava bean, 11S, and 7S at pH 2 and pH 7.

Sample	pH	K (Pa.s)	n
Fava bean	2	0.217 ± 0.016	−0.770 ± 0.021
	7	0.009 ± 0.001	−0.293 ± 0.081
11S	2	0.087 ± 0.063	−0.727 ± 0.005
	7	0.011 ± 0.047	−0.360 ± 0.129
7S	2	0.076 ± 0.012	−0.720 ± 0.029
	7	0.013 ± 0.007	−0.424 ± 0.014

## Data Availability

Data is contained within the article or Appendix A.

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
