# Peer review of "Protein Nanofibrils from Fava Bean and Its Major Storage Proteins: Formation and Ability to Generate and Stabilise Foams"

_foods, 2023, doi:10.3390/foods12030521_

Round 1
Reviewer 1 Report (Previous Reviewer 1)
The authors create nanofibrils from fava proteins and studied the ability of different fava proteins to form the fibrils. In general, the article is well-written and discussed, however, there are a few points that need clarification.
The authors could make clear in the introduction what is the difference between fibrillated protein and non-fibrillated, can these characteristics be tailored according to the processing method?
Very high salt concentrations were used to separate the 7S and 11S and it seems there was no desalting step. Why wasn’t desalting carried out?
Line 98: Add info about the centrifuge used in the study
Section 2.1.2. : How were the conditions to obtain 7 and 11S defined, were they taken from the literature or from the research group`?
In the conclusion, the authors should emphasize how the results of this study can contribute to food science.
Author Response
Reviewer 1
The authors create nanofibrils from fava proteins and studied the ability of different fava proteins to form the fibrils. In general, the article is well-written and discussed, however, there are a few points that need clarification.
The authors could make clear in the introduction what is the difference between fibrillated protein and non-fibrillated, can these characteristics be tailored according to the processing method?
A: Thank you for your comment. A clarification of what the differences in preparation were between fibrillated and non-fibrillated protein can now be found on lines 83-84.
“The only difference in the preparation of the samples was that the fibrillated proteins had been heated at 85 °C for 24h.”
Very high salt concentrations were used to separate the 7S and 11S and it seems there was no desalting step. Why wasn’t desalting carried out?
A: 7S and 11S proteins were obtained at 150 mM and 300 mM final salt concentrations. For each experiment, the proteins were prepared by diluting approximately 100 or 1000 times to 10 or 1 mg/ml. We considered the final salt concentration to be low at this point and therefore no desalting step was carried out
Line 98: Add info about the centrifuge used in the study
A: The centrifuge model has now been added on lines 98-99.
“The mixture was then stirred at room temperature (20 °C, 1 h), followed by centrifugation (Sorval Lynx 4000l, Thermo Scientific) at 3700 g for 30 min”
Section 2.1.2. : How were the conditions to obtain 7 and 11S defined, were they taken from the literature or from the research group`?
A: We prepared 11S and 7S using a similar method described in Suchkov et. Al 1990 with some modification. This information has been now added to the manuscript see line 107.
“The 11S (legumin) and 7S (vicilin) fractions were extracted from whole fava bean using a similar protocol described by Suchkov et al. (1990) [16], with some modifications (Figure 1A). “
In the conclusion, the authors should emphasize how the results of this study can contribute to food science.
A: Thank you for your comment have now updated the conclusions, emphasizing how our results can contribute to food science see lines 455-457. “The results from this studies have generated a greater understanding of how plant-based PNFs with different morphology can contribute to stabilise future food applications”
Reviewer 2 Report (Previous Reviewer 2)
I would like to thank the authors for considering my comments and processing these in the main text.
Most comments are answered accordingly. A few minor points remain:
Comment 9:
Reviewer comment in first round: Sample preparation: can the authors also mention how the samples were prepared in general? So how long are the samples stirred? And at what pH?
authors answer in revision: Information about PNFs sample preparation can be found on lines 135-140. Information on how the foams were prepared can be found on lines 186-191.
Reviewer comment round 2: indeed info about PNF sample preparation is present in line 135-140. But what is missing is if these samples are directly used in the other experiments. If yes, please state specifically if these samples are used directly after preparation. Also, it would be great to mention how long the samples are used after preparation and how they are stored in this period.
Comment 10: great to hear that the authors use a calibration curve. In that case, the authors can relate the peaks of the chromatogram to specific protein molecular weights. Can the authors do this for their graphs and add this to the manuscript? Giving specific Mw's would really help the reader a lot.
Comment 11: can the authors still add info of the cantilevers? Spring contant, etc. Also what force and amplitude is used during the measurement? Please add this info. This standard info should be provided when performing AFM.
Comment 12: please mention the PROBE connected to the turrax. The Turrax 25 is the name of the machine, not the probe. Sample: https://www.analytics-shop.com/gb/ik0001713300-gb.html
The type of probe (diameter, type of cavities) has huge impact on foaming properties.
Comment 13: the appropriate location for an appropriate explanation would be in the R&D. In paragraph line 255-268. Could you also address this in this section?
Self citation: the selfcitation is rather high, it's 25% atm. This should be <20%.
Please change accordingly.
Author Response
Reviewer 2
I would like to thank the authors for considering my comments and processing these in the main text.
Most comments are answered accordingly. A few minor points remain:
Comment 9:
Reviewer comment in first round: Sample preparation: can the authors also mention how the samples were prepared in general? So how long are the samples stirred? And at what pH?
authors answer in revision: Information about PNFs sample preparation can be found on lines 135-140. Information on how the foams were prepared can be found on lines 186-191.
Reviewer comment round 2: indeed info about PNF sample preparation is present in line 135-140. But what is missing is if these samples are directly used in the other experiments. If yes, please state specifically if these samples are used directly after preparation. Also, it would be great to mention how long the samples are used after preparation and how they are stored in this period.
A: Thank you for this comment. Have now added some clarification on how the samples was stored and for how long before further analysis and experiments were conducted see lines 142-145.
“After heating the fibrillated samples were cooled on ice and then stored at 4 °C. PNFs detection was made directly after the fibrillated samples was cooled and the foaming experiments was conducted within a week after preparation. “
Comment 10: great to hear that the authors use a calibration curve. In that case, the authors can relate the peaks of the chromatogram to specific protein molecular weights. Can the authors do this for their graphs and add this to the manuscript? Giving specific Mw's would really help the reader a lot.
A: We have used gel filtration calibration kit in which the proteins Ferritin (440 KD), Aldolase (158 KD), Conalbumin (75 KD), and ovalbumin (44 KD) were included. The corresponding molecular weights of 11S and 7S have been added next to the elution volumes on the graph now see figure 1b.
Comment 11: can the authors still add info of the cantilevers? Spring contant, etc. Also what force and amplitude is used during the measurement? Please add this info. This standard info should be provided when performing AFM.
A: Have now added some clarification about tip radius, spring constant and peak force amplitude see lines 163-165.
“FastScan B cantilevers (Bruker, tip radius = 5 nm, spring constant = 2 N/m, peak force amplitude = 150 nm) were used for the experiments, and the micrographs were analysed with Gwyddion 2.48 (http://gwyddion.net/).”
Comment 12: please mention the PROBE connected to the turrax. The Turrax 25 is the name of the machine, not the probe. Sample: https://www.analytics-shop.com/gb/ik0001713300-gb.html
The type of probe (diameter, type of cavities) has huge impact on foaming properties.
A: The probe that was used was an S25N (IKA ®) this information has now been added on line 194.
“The probe S25N (IKA ®) was immersed in the solution, the homogeniser (IKA ® Ultra Turrax T25) was started and the speed was gradually increased from 8000 rpm to 13500 rpm.”
Comment 13: the appropriate location for an appropriate explanation would be in the R&D. In paragraph line 255-268. Could you also address this in this section?
A: I´m so sorry but we don´t understand this question. The section you refer to is already in the result and discussion part. Could you please clarify if there is something we have missed to add in this section?
Self citation: the selfcitation is rather high, it's 25% atm. This should be <20%.
Please change accordingly.
A: We have now rewritten some part to get down the self-citations.
Reviewer 3 Report (Previous Reviewer 3)
The resubmitted paper has not been greatly improved according to the reviewers' suggestions and comments.
Author Response
The resubmitted paper has not been greatly improved according to the reviewers' suggestions and comments.
A: Hope that the updated manuscript and the answers to the other reviewers are to your satisfaction. If not please clarify what information you are missing and we will include this in the second revision round.
Reviewer 4 Report (Previous Reviewer 4)
1、Line 54-56 ,“In an early study by our group, we demonstrated that it is possible to produce PNFs from 54a broad range of plant-based proteins by exposing the protein to an acid environment and 55 heat ”, As you described, the preparation method of PNF will cause structural changes in proteins to a certain extent, do you think that the PNF prepared by this method will cause safety hazards as food texture adjustment? How is the safety of PNF research evaluated in food applications? Does the concentration of PNF involved in your study meet this safe concentration?
2、Some figures do not have symbols in the data that are not statistically processed, please add.( Figure 2 and Figure 5)
3、Please include in your conclusions the outlook for the important results of your research and possible areas of application.
Author Response
Reviewer 4
1、Line 54-56 ,“In an early study by our group, we demonstrated that it is possible to produce PNFs from 54a broad range of plant-based proteins by exposing the protein to an acid environment and 55 heat ”, As you described, the preparation method of PNF will cause structural changes in proteins to a certain extent, do you think that the PNF prepared by this method will cause safety hazards as food texture adjustment? How is the safety of PNF research evaluated in food applications? Does the concentration of PNF involved in your study meet this safe concentration?
A: There are very few studies today investing in the safety of PNFs. There is one in vitro study made by Lassé et al (2016) on PNFs made from whey, soybean, kidney bean, and egg with protein where they found that a concentration of PNFs up to 0.25 mg/ mL exerted low or non-toxicity on human cell lines (Caco-2 and Hec-1). To our knowledge, there is only one in vivo study made on PNFs. Shen et al 2017 showed that whey-based PNFs fortified with iron nanoparticles had good digestibility and bioavailability when investigated in animal models.
In this study, we are focusing on the functionality of plant-based PNFs but further research needs to be done to ensure the safety aspect of PNFs. We also have an ongoing study looking to the safety aspect of PNFs from whey and several plant-based sources so hopefully, we can contribute to more knowledge in this field in close future.
Lassé, M., Ulluwishewa, D., Healy, J., Thompson, D., Miller, A., Roy, N., Chitcholtan, K. and Gerrard, J.A. (2016) ‘Evaluation of protease resistance and toxicity of amyloid-like food fibrils from whey, soy, kidney bean, and egg white’, Food Chemistry, 192, pp. 491–498. Available at: https://doi.org/10.1016/j.foodchem.2015.07.044.
Shen, Y., Posavec, L., Bolisetty, S., Hilty, F.M., Nyström, G., Kohlbrecher, J., Hilbe, M., Rossi, A., Baumgartner, J., Zimmermann, M.B. and Mezzenga, R. (2017) ‘Amyloid fibril systems reduce, stabilize and deliver bioavailable nanosized iron’, Nature Nanotechnology, 12(7), pp. 642–647. Available at: https://doi.org/10.1038/nnano.2017.58.
2、Some figures do not have symbols in the data that are not statistically processed, please add.( Figure 2 and Figure 5)
A: The error bars refer to the standard deviation. This has now been added to the figure legend for Figures 2, 4, and 5.
3、Please include in your conclusions the outlook for the important results of your research and possible areas of application.
Thank you for your comment have now updated the conclusions, emphasizing how our results can contribute to food science see lines 455-457.
“The results from this studies have generated a greater understanding of how plant-based PNFs with different morphology can contribute to stabilise future food applications”
Round 2
Reviewer 2 Report (Previous Reviewer 2)
The paper can be accepted as such
Author Response
Thank you for taking the time to review our manuscript, reviewer 3 and 4 had some minor comments so the manuscript has been updated according to their comments.
Reviewer 3 Report (Previous Reviewer 3)
In the introduction part, the authors should provide more information about the previously reported references about the protein nanofibril and protein nanofibrils from fava bean. The aim and novelty of their research are not well described.
More structure and property information of 11S and 7S are needed to be provided, and the difference between them and why author choose them for investigation should also be addressed.
The authors should compare their results with the other reported results about the 11S, 7S and fava bean or other protein nanofibrils.
The mechanism for the difference of the foaming properties of these protein nanofibrils should be discussed more, better based on some results from the interfacial structure and property characterizations.
It seems that the information is not enough for this paper.
Author Response
Reviewer 3
In the introduction part, the authors should provide more information about the previously reported references about the protein nanofibril and protein nanofibrils from fava bean. The aim and novelty of their research are not well described.
A: Thank you for this comment. We have now added more information about our previous paper about plant-based protein nanofibrils where fava bean was included. See lines 54-66
“In an early study by our group, we demonstrated that it is possible to produce PNFs from a broad range of plant-based proteins by exposing the protein to an acid environment and heat [10]. One of the plant-based PNFs characterised was made from a whole protein isolate extracted from locally produced fava beans. When visualising the PNFs with atomic force microscopy (AFM) the result showed that depending on the protein source the PNFs varied in morphology (being either curved or straight) and in length (~220-910 nm). The plant-based PNFs were compared to a well-studied animal-based protein (whey). Here the result showed that whey formed considerably longer PNFs (several µm) than any of the PNF from plant-based sources. We hypothesized that whey proteins superiority could be due to that the protein isolate was very pure and consisted of a majority of two very small proteins (β-lactoglobulin and α-lactalbumin), with a molecular weight of 18.3 and 14.2 kDa [11]. The plant-based globulins investigated were approximately 10-20 times larger [12–16] than the major protein found in whey proteins.”
More structure and property information of 11S and 7S are needed to be provided, and the difference between them and why author choose them for investigation should also be addressed.
A: This has now been included in the introduction see lines 66-72.
“The fava bean protein used in this previous study was extracted in-house with isoelectric precipitation generating a protein consisting of a mixture of the larger 11S globulins of ~353 kDa and the smaller 7S globulins with a molecular weight of ~150kDa [12]. By separating the two globular fractions and comparing the PNF forming ability with the whole fava bean protein isolate further information can be revealed about the impact of protein size on PNF formation.”
The authors should compare their results with the other reported results about the 11S, 7S and fava bean or other protein nanofibrils.
A: We believe that we already have compared our results with early reported proteins and protein nanofibrils from other sources, see highlighted below.
Lines 282-284
“A similar lack of increase in fluorescence after fibrillation for fraction 11S has been observed previously by our research group for protein isolates extracted from oat and rapeseed, which mainly consist of 12S globulins [12,16, 17].”
Lines 297-306
“Based on the results from the ThT assay (Figure 2) and AFM analysis (Figure 3), it can be concluded that 7S globulins from fava bean are superior to 11S globulins in forming PNFs. This agrees with earlier findings on PNF formation from the 7S and 11S globular fractions in soybean [25]. Using several different methods, such as ThT assay, Congo read spectroscopy assay, circular dichroism spectroscopy and SDS-gel separation that study confirmed that when heated at pH 2, the protein from the soybean 7S fraction formed PNFs more readily than the 11S fraction under the same conditions. The reason for 7S being superior to 11S globulins in forming PNFs is not fully understood. Tang and Wang speculated that it might be because 7S contains more charged amino acids, which hydrolyses into peptides more easily at low pH and high temperature [25]”
Lines 316-324
“When pH of the fibrillated samples was increased to 7, the fibrillary structure degraded in all samples (Figure 3D-E). For fibrillated fava bean protein, the samples mainly contained globular aggregates (Figure 3D). Both 11S and 7S seemed to have some shorter fibrils, which grouped into larger aggregates in some cases (Figure 3E-F). These results show that the PNFs from whole fava bean and fractions 11S and 7S are less stable at pH 7 than at pH 2. Fibrillated proteins from whey, soybean and mung bean have also been shown to be less stable at pH 7 [20, 21, 14]. However the PNFs from those sources did not fragment as much as the fava bean 11S and 7S PNFs, so our results in this regard were unexpected.”
Lines 342-358
“This was probably because the PNFs were degraded to smaller particles (see Figure 3D-F). In an earlier study by our group investigating the viscosity of PNFs generated from mung bean protein [20], it was found that those PNFs had a distinctly different morphology than the PNFs from fava bean-based proteins in the present study, with curved structure instead of straight. The apparent viscosity of PNFs generated from mung bean (Figure S1 in Supplementary Material (SM)) under the same conditions as applied here (10 mg/mL, pH 2) showed that the fibrillated mung bean protein had around 1.7-6.3 times higher apparent viscosity than the fibrillated protein from fava bean and its globulin fractions. This was also confirmed with that the fibrillated mung bean samples had a higher consistency index (Table S1) than the fibrillated fava bean, 11S and 7S (0.386 vs 0.217, 0.087 and 0.076 respectively) at pH 2. Similar differences in viscosity profile have been reported for curly and straight PNFs made from β-lactoglobulin, where the curved PNFs had higher viscosity than the straight PNFs at the same concentration and conditions [29]. Z potential measurements of both fibrillated and non-fibrillated proteins varied between 20.3-25.5 mV at pH 2, - 5.5 and -26.9 mV at pH 5, and -13.8 and -21.3 mV at pH 7 (Figure 4B). The fibrillated samples had similar Z potential at respective pH, as earlier observed for fibrillated whey protein [30]”
Lines 425-427
“Additionally, it cannot be excluded that the peptides in the fibrillated fava bean, 11S, and 7S samples had an impact on the superior foaming properties at the higher concentration (10 mg/mL), as early observed for fibrillated samples from soy 11S [19].”
Lines 430-436
“To summarise, based on the results from this study, our previous study about curved PNFs from mung bean and the study conducted by Wan et al. (2021) about PNF from soy 11S [19], it appears that no general conclusions can be drawn about plant-based PNFs foam stabilisation ability. PNFs will have different morphology (straight/curved) depending on protein sources. Our findings indicated that curved PNFs are superior in their foam stabilising ability because of their ability to form a more viscous sample.”
Lines 437-448
” The non-fibrillated proteins had approximately the same foaming capacity and foam stability/half-life at low concentrations (Figure 6A-C) as at high concentrations (Figure 5A-C). Similarly, a previous study investigating the foaming properties of a fava bean protein isolate at pH 7 and concentrations from 0.1-3% (w/v) observed that higher concentrations did not give any significant improvement in foaming capacity or stability [32]. In contrast, the foams generated from non-fibrillated protein in the present study had higher foaming capacity. This might be due to the differences in the method used for foam formation or sample preparation. The only difference in treatment between the non-fibrillated and fibrillated proteins examined in this study was that the non-fibrillated proteins were not heated. Initial adjustment of the pH to 2 might have caused partial unfolding of the native protein structure, generating smaller and more flexible structures and exposing hydrophobic sites that could help stabilise the air/liquid interface [33].”
The mechanism for the difference of the foaming properties of these protein nanofibrils should be discussed more, better based on some results from the interfacial structure and property characterizations.
A: We have discussed the possible mechanism for the foaming properties of the different morphological protein nanofibrils (see lines 415-436). Our purpose with this paper was not to investigate the interfacial properties of the foams made from both fibrillated and non-fibrillated protein rather more to investigate if PNFs could be used for stabilising foams and if PNFs concentration and morphology has any impact on these mechanisms, thus we believe that the discussion should be sufficient. We agree that it would be interesting to further investigate the interfacial structure of these foams but the addition of that kind of information needs quite a lot of additional experiments something that we can consider for a future separate manuscript.
“However, it is still not fully understood how curved PNFs contribute to stabilising the foams at low concentrations. A possible explanation is that curved PNFs increase the bulk viscosity in the solution’s continuous phase, and thereby sterically hinder bubble rupture. At pH close to the isoelectric point, the viscosity increased even more due to aggregation of the PNFs, which probably explains why the foam made from pure mung bean fibrils was most stable at pH 5 [20]. The fibrillated protein from fava bean, 11S and 7S did not form stable foams at this low concentration (1 mg/mL), even at pH 5 (Figure S2 in SM). The less stable foams probably developed because the straight PNFs generated from fava bean and its major globulin fractions (11S and 7S) were not sufficiently viscous to help stabilise the continuous phase between the bubbles at these low concentrations. Additionally, it cannot be excluded that the peptides in the fibrillated fava bean, 11S, and 7S samples had an impact on the superior foaming properties at the higher concentration (10 mg/mL), as early observed for fibrillated samples from soy 11S [19]. The PNFs made from fava bean and its globular fraction might be too inflexible to stabilise the interfacial film of the air bubbles. At the lower concentration (1 mg/mL) the peptides might be too diluted to maintain the stabilising effect observed at higher concentrations. To summarise, based on the results from this study, our previous study about curved PNFs from mung bean and the study conducted by Wan et al. (2021) about PNF from soy 11S [19], it appears that no general conclusions can be drawn about plant-based PNFs foam stabilisation ability. PNFs will have different morphology (straight/curved) depending on protein sources. Our findings indicated that curved PNFs are superior in their foam stabilising ability because of their ability to form a more viscous sample. “
It seems that the information is not enough for this paper.
Reviewer 4 Report (Previous Reviewer 4)
1、Please add a research introduction to PNFs in the preface, which can be an example of the literature you supplemented, and make an appropriate summary.
2、Some figures do not have symbols in the data that are not statistically processed, please add.( Figure 5b and Figure 6a,6b)
3、Line453-455,Results from this study and our earlier study on foaming 453 properties of mung bean based PNFs show that curved PNFs are superior to straight semi-454 flexible PNFs in their ability to stabilise foams at low concentrations. Please limit the low concentration.
4、Please emphasizing how your results can contribute to food science,add a few more words.
Author Response
Reviewer 4
1、Please add a research introduction to PNFs in the preface, which can be an example of the literature you supplemented, and make an appropriate summary.
A: Thank you for your comment. We have now added more information about plant-based protein nanofibrils in the introduction see lines 54-66
“In an early study by our group, we demonstrated that it is possible to produce PNFs from a broad range of plant-based proteins by exposing the protein to an acid environment and heat [10]. One of the plant-based PNFs characterised was made from a whole protein isolate extracted from locally produced fava beans. When visualising the PNFs with atomic force microscopy (AFM) the result showed that depending on the protein source the PNFs varied in morphology (being either curved or straight) and in length (~220-910 nm). The plant-based PNFs were compared to a well-studied animal-based protein (whey). Here the result showed that whey formed considerably longer PNFs (several µm) than any of the PNF from plant-based sources. We hypothesized that whey proteins superiority could be due to that the protein isolate was very pure and consisted of a majority of two very small proteins (β-lactoglobulin and α-lactalbumin), with a molecular weight of 18.3 and 14.2 kDa [11]. The plant-based globulins investigated were approximately 10-20 times larger [12–16] than the major protein found in whey proteins.”
2、Some figures do not have symbols in the data that are not statistically processed, please add.( Figure 5b and Figure 6a,6b)
A: Figure 5 b has now been updated with error bars. Due to limited material we were not able to run analyses in duplicate for all samples at low concentration. However, our pilot study (data not shown) demonstrated a similar trend, which confirm the reliability of the present results. Thus, Figure 6a and 6b do not have any error bars.
3、Line 453-455,Results from this study and our earlier study on foaming 453 properties of mung bean based PNFs show that curved PNFs are superior to straight semi-454 flexible PNFs in their ability to stabilise foams at low concentrations. Please limit the low concentration.
A: Thank you for your comment, have now updated the sentence see lines: 464-466.
“Results from this study and our earlier study on foaming properties of mung bean based PNFs show that curved PNFs are superior to straight semi-flexible PNFs in their ability to stabilise foams at low concentrations (1 mg/mL).”
4、Please emphasizing how your results can contribute to food science,add a few more words.
A: Have now elaborated on how the results from these studies can contribute to food science see lines: 469-475.
“Introduction of stable air bubbles in food products are important for the appearance and mouthfeel of many food applications. Today animal proteins are superior to plant-based proteins in the aspect of creating stable bubbles in food. Here we show that by reconstructing the plant proteins into PNFs the ability to stabilise air bubbles increases dramatically, generating new insight into how plant protein can be used to create suitable food applications”
This manuscript is a resubmission of an earlier submission. The following is a list of the peer review reports and author responses from that submission.
Round 1
Reviewer 1 Report
In general, the article is very well written and discussed. However, the article lacks further analysis that are fundamental to fully understanding the stabilization mechanism of the foams. I suggest further analyses of, zeta potential, surface tension and surface hydrophobicity are carried out. In addition, I have another concern on the products feasibility, thinking in terms of application, is it viable to leave a dispersion heating at 80 C for 24 hours. What are the author thoughts on that?
The discussion on viscosity can be improved and data acquired can be discussed further, for example was there an initial tension? How about adjusted according to the power law model or Herschel-Bulkley model to determine the flow consistency index and n is and the flow behavior index? Moreover, the most appropriate terminology is apparent viscosity.
Other minor comments:
Line 66: Define VWR
Line 79: Why was pH adjusted to 4 before freeze-drying? There are many studies reporting that adjusting to neutro pH is better.
Line 131: At what concentration was the measurements carried out? Include the information
Eq 1 and 2: Fix: add parentheses on the fraction as the way it is written one can make confusion
Line 267: lower case the s
Line 286: There is no title for the figure
Reviewer 2 Report
The authors create nanofibrils from fava proteins and studied the ability of different fava proteins to form the fibrils. In addition, the foaming properties were studied. The novel part is the studying of the 7S and 11S proteins. The article is moderately-well written. Organization & use of English is appropriate. Quite some points should be better addressed, see the comments below. However, there are two major concerns, which MUST be addressed.
1. 1. The 7S & 11S proteins are purified using the method with high NaCl concentrations. This is a common and effective method, but it seems the authors did not desalt the samples. The final sedimentation step is sometimes not sufficient to desalt. Also in future steps, there is no salt removal. This is a concern, because the ionic strength has an immense impact on foaming properties. Can the authors give the protein contents of the 7S & 11S protein extract powders (before fibril formation) and also after fibril formation? Also, please check the ionic strength or conductivity of the prepared solutions and report this in the manuscript. If the ionic strength of the 7S & 11S protein (fibrills) are elavated, additional tests are required to check the impact of the higher ionic strength.
2. 2.The second point is the described mechanisms. This point is extensively addressed as the last comment (see below). This reviewer is not convided that the air bubbles & foams are stabilized by the fibrils. Previous works have shown that the free peptides (that are not in the fibrils) dominate the interface & foam stabilizing properties. The fibrils itself are barely surface-active and are very poor foam stabilizers. Check the full comment below for more explanation and points. Separating the free peptides + fibrils & studying their foaming properties would largely increase the quality of this article. If the authors are not able/willing to do this experiment, the foaming part should be rewritten carefully.
The recommendation is a major revision, with a great emphasis on the two previously mentioned points.
Line 34: the authors mention texture here. Also in line 36, and also in the abstract. I understand that texture in this context is used as a general term to define (food) structures that are created with ingredients. However, texture is also widely used to address (semi-)solid materials, such as gels and extruded materials. I would highly recommend to use another term for texture here, as using texture in a foam context might be confusing. Perhaps the authors can talk about functionality? For instance, in line 36: Texture is essential for generating stable food foams. In this context, the word texture may be highly confusing.
Line 37: The authors mention that proteins can form thin films that capture the air bubble. Here, the authors should use the correct terminology. The air bubbles are captured by an protein interfacial film. The thin film is the liquid layer between two interfaces.
Line 46: perhaps the authors can spend a few more lines, why the proteins hydrolyze at low pH + heat. And why the peptides would form fibrils at all.
Line 62: It would be great to end the introduction with a perspective how the findings of this work can be applied in food science.
Material & methods: did the authors determine protein content just by weighing? Or did the authors did a more accurate method, such as dumas or kehldahl? Can the method be mentioned? And is the protein content mentioned in later points in the text? Related to this point: the concentration mentioned in line 104, is the final conc. Of 50 mg/ml based on protein on dry matter? This reviewer has a major concern about this point. High salt concentrations were used to separate the 7S and 11S. There seems to be no diafiltration/dialysis step. So extremely high salt concentrations are present, which will also impact the foaming properties. So, it is key to show the protein concentrations, and also measure the conductivity of the samples for the foaming experiments.
Sample preparation: can the authors also mention how the samples were prepared in general? So how long are the samples stirred? And at what pH?
2.2 SEC: can the authors give more info: what was the flow rate. And what was the wavelength of the detector? And what machine did you use? Did the authors run a calibration curve with pure proteins with known sizes?
2.5. AFM: also here, please provide more info. What dimensions did you measure? How many replicates on how many locations? What were the settings of the measurement? Spring constant, force, amplitude, etc.
2.7 foam: can the authors mention which probe? There are many Ultra Turrax probes.
Line 206: The authors mention that the ThT assay is used, as the dye binds to B-sheet-rich structures. Can the authors address in the text, why more B-sheet structures are expected for after fibril formation?
Figure 1: can the authors explain the peak at 100 mL?
Major concern foaming properties: this reviewer is not convinced that the air bubbles & foams are stabilized by the fibrils. The authors cite two previous works on soybean protein fibres (ref 8 and 24) by Wan et al. These work demonstrated that only a part of the peptides forms the fibrils. Wan et al showed that these peptides were highly surface active and dominate the interface- and foam-stabilising properties. I am convinced that this is also the case here. The peptides will give a higher foam capacity, and smaller air bubbles. Wan et al. showed that the fibrils were actually very bad foam stabilizers by itself. The fibrils might act as a blockage in the thin films (between air bubbles) to reduce the drainage, thus slightly increase foam stability (this phenomenon is called pinning). But this effect was not that large, as shown by Wan et al. The impact of the free proteins in aggregated systems has been studied before, and nearly all cases, the free proteins seem to dominate the interface & foam stabilising properties. So, the free peptides will play a major role in foam stabilisation, and the fibrils could help increase the stability.
This also explains why the foaming properties were much better at a higher concentration. At a high concentration: there is an absolute higher peptide content. Image that 20% of the total mixture are free peptides. At a 1 mg/ml solution, you would be comparing 0.2 mg/ml free peptides against 1 mg/ml non-fibrillated proteins. At 10 mg/ml, you would be comparing 2 mg/ml free peptides against 10 mg/ml non-fibrillated proteins. Sufficient free peptides are present in this case to give stable foams.
This explanation should be incorporated in the manuscript. Separating the fibrils from the peptides, and performing foam experiments would largely increase the quality of this manuscript. If the authors are not able or willing to perform the experiment, I would recommend to rewrite the foaming part carefully. The main message: the protein fibril solution is a good foaming agent, but the fibrils themselves are probably not very good foaming agents.
Reviewer 3 Report
1. In the introduction part, the authors should provide a summary about the references about the foams properties of protein nanofibrils (PNFs). The novelty of their research should be well addressed.
2. In this paper, the authors investigated three proteins. The authors should give more information about the size, weight fraction, structure and properties of these three proteins. In this paper, the author said that "Most PNFs in the fibrillated 7S samples were short (~100-200 nm), but with some longer 229 fibrils co-existing in the same samples. The longest fibril measured was 2.2 µm." How do the authors know that these PNFs are in nanometer size, diameter?
3. The authors should provide a deep discussion about the reason and mechanism for the difference between the foaming properties of these different proteins. The discussion looks simple and unclear. The readers can not get attractive points. The relationship between structures and properties should be well discussed.
4. In the results and discussion part, the authors should provide some comparison of the foam properties between their fava bean protein nanofibrils and other protein nanofibrils reported before.
Reviewer 4 Report
1、Are there any studies on protein nanofibrils from fava beans? These should be highlighted in the introduction.
2、What are the current methods for preparing protein nanofibrils?
3、In addition to your team's method, what other methods can be used to prepare? These should be properly explained in the introduction.
4、Lin98-101:The introduction of the method is not comprehensive, please explain the specific content clearly.
5、What is the protein content of faba bean protein and its storage protein component protein(11S and 7S)? These should be introduced in the source material.
6、Figure 1 in the attached material and Figure 5 in the document (lin286-287) into a single figure.
7、Figure 5 in the article seems to have no title, please add.
8、The content of the reference is incomplete, please add it.(Line401-402)
9、The results and analysis in the article are a little simpler and can be supplemented appropriately.